**Data Availability Statement:** All relevant data are within the paper and its Supporting information files.

**Funding:** UV is funded by Epilepsy Research UK (P1906) and the Medical Research Council (MR/

# Alternating hemiplegia of childhood: An electroclinical study of sleep and hemiplegia

Josephine Poole[1,2]*, Sara Zagaglia[1,2], Rita Demurtas[3], Fiona Farrell[1,2], Matthew C. Walker[1], Sanjay M. Sisodiya[1,2], Simona Balestrini[1,2,4]°, Umesh Vivekananda[1]° 

1 Department of Clinical and Experimental Epilepsy, UCL Queen Square Institute of Neurology, London, United Kingdom, 2 Chalfont Centre for Epilepsy, Chalfont St. Peter, United Kingdom, 3 Neurology and Stroke Unit, Azienda Ospedaliera SS Antonio e Biagio e Cesare Arrigo, Alessandria, Italy, 4 Children's Hospital A. Meyer, University of Florence, Paediatric Neurology, Neuroscience Department, Florence, Italy

☉ These authors contributed equally to this work.
* josephine.poole.19@ucl.ac.uk

## Abstract

### Objective

Alternating Hemiplegia of Childhood (AHC) is characterised by paroxysmal hemiplegic episodes and seizures. Remission of hemiplegia upon sleep is a clinical diagnostic feature of AHC. We investigated whether: 1) Hemiplegic events are associated with spectral EEG changes 2) Sleep in AHC is associated with clinical or EEG spectral features that may explain its restorative effect.

### Methods

We retrospectively performed EEG spectral analysis in five adults with AHC and twelve age-/gender-matched epilepsy controls. Five-minute epochs of hemiplegic episodes and ten-minute epochs of four sleep stages were selected from video-EEGs. Arousals were counted per hour of sleep.

### Results

We found 1) hemispheric differences in pre-ictal and ictal spectral power ($p = 0.034$), during AHC hemiplegic episodes 2) 22% reduced beta power ($p = 0.017$) and 26% increased delta power ($p = 0.025$) during wakefulness in AHC versus controls. There were 98% more arousals in the AHC group versus controls ($p = 0.0003$).

### Conclusions

There are hemispheric differences in spectral power preceding hemiplegic episodes in adults with AHC, and sleep is disrupted.

### Significance

Spectral EEG changes may be a potential predictive tool for AHC hemiplegic episodes. Significantly disrupted sleep is a feature of AHC.

T033150/1). The funders had no role in study design, data collection and analysis, decision to publish, or preparation of this manuscript.

**Competing interests:** The authors have declared that no competing interests exist.

## Introduction

Alternating hemiplegia of childhood (AHC) is a rare and severe neurological disorder with a prevalence of 1:100,000 to 1:1,000,000 [1,2]. The clinical phenotype is characterised by developmental delay, recurrent, transient episodes of unilateral or bilateral paralysis, epileptic seizures, abnormal eye movements, transient autonomic dysfunction, cardiac abnormalities, dystonic posturing, and tonic attacks [3,4]. Onset of AHC typically occurs within the first 18 months of life and has been associated with mutations, usually occurring *de novo*, in the *ATP1A3* gene, which encodes the alpha3 subunit of the $Na^+/K^+$-ATPase transporter [5]. Epilepsy is comorbid in 50–60% of patients with AHC, often drug-refractory, and frequently involves recurrent episodes of status epilepticus [5,6].

One of the cardinal diagnostic criteria of AHC is the remission of hemiplegia and other paroxysmal events, but not seizures, with sleep, and their potential reappearance shortly after waking [7]. Recent work has demonstrated altered sleep-wake patterns in children with AHC, with polysomnography recordings revealing the presence of frequent apnoeas and arousals in a cohort of 22 children [8]. However, it is not clear if these features persist into adulthood. Moreover, the neurophysiological mechanisms underlying the 'therapeutic' effect of sleep in AHC remain unknown. Hemiplegic episodes (HE) have been suggested to be due to enhanced cortical spreading depression, secondary to extracellular potassium dysregulation. If this is so, then changes in the spectral content of the EEG would be expected to occur during HE. Scalp EEG has been recorded during HE and no associated epileptiform EEG changes were noted [6]. However, more detailed analysis of the EEG, in particular frequency domain analysis during HE, was not undertaken.

In this retrospective study, we used clinical phenotyping, video-EEG data, and spectral analysis of sleep-, wake-, and ictal-EEG to investigate the following hypotheses: 1) whether HEs are associated with EEG changes in the frequency spectrum domain and 2. whether sleep in AHC is significantly different to controls with well-controlled epilepsy, both clinically (in terms of arousals) and electrographically (in terms of spectral power).

## Methods

### Patients and study design

Approval was granted by UCLH research ethics committee (11/LO/2016) and either written informed consent, or written assent given by parents or legal guardians, was obtained.

The inclusion criteria were a diagnosis of AHC according to the Aicardi clinical criteria [7], and prolonged video-EEG study available (with at least one hour of wake and two hours of sleep). Neither the AHC group or epilepsy control group had patients with a co-existing sleep disorder (e.g. obstructive sleep apnoea, parasomnias). Retrospective clinical phenotyping and spectral analysis of sleep, wake, and ictal EEG was carried out in five adult patients (all female) with AHC who underwent clinical video-EEG telemetry monitoring between 2009–2018 at the Sir William Gowers Centre, part of University College London Hospitals (UCLH). Two telemetry recordings (5 years apart) were available for Patient 2; data from both were included in the analysis due to phenotypic variability in AHC (adjusted n = 6). Age- and sex-matched control EEG data (n = 12) were retrospectively obtained from a consecutive cohort of patients with well-controlled epilepsy (seizure-free for at least six months prior to video telemetry (VT) and no epileptiform abnormality evident during video-EEG) (S1 Table).

## Clinical phenotyping

All available clinical notes for each patient, including reports from carers, were reviewed for: genetic data; epileptic/seizure history (where applicable); HE; non-epileptic paroxysmal episodes; non-paroxysmal features, such as developmental delay and intellectual disability; comorbidities; and neuroimaging data (where available).

## Video telemetry

Prolonged clinical video-EEG (24–96 hours) data and reports were reviewed retrospectively. EEG systems and channel set ups varied between patients, so recordings from a set of common electrodes defined in the international 10–20 system (Fp1, Fp2, F7, F3, Fz, F4, F8, T3, T4, T5, T6, C3, Cz, C4, P3, Pz, P4, O1 and O2) were analysed in average reference. EEG sampling rate for all studies was 512Hz. All EEG was manually reviewed by a researcher and a trained neurophysiologist, with any epochs containing clinical or electrographic events, such as HEs, seizures, or interictal spikes excluded from the EEG analysis.

## Spectral analysis

Prior to analysis, data files were pre-processed in EEGLAB [9]. EEG data from the 19 common electrodes were filtered (FIR filter 1–45 Hz) and resampled to 250 Hz prior to visual inspection for identification of EEG channels and epochs that were marred by artefact; these were then excluded from analysis. Independent Component Analysis (EEGLAB SOBI algorithm) was applied to the data to remove eye blink, heart, muscle, and line noise components. The resultant data were divided into 5-second contiguous epochs. As resting data was used across patient groups, the derivative of the raw signal was computed to remove 1/frequency effects. Spectral power for each channel was then computed in 1Hz bins using a fast Fourier transform method (FieldTrip toolbox) and z-scored within patient for the different time periods. For the sleep analyses, relative power in the delta (0–<4 Hz), theta (4–7 Hz), alpha (8–12) and beta (13–25) bandwidths for each channel were calculated.

For the HE analyses, mean z-scored power was calculated at sensor level for left hemisphere (Fp1, F7, F3, T3, T5, C3, P3, O1) and right hemisphere (O2 Fp2, F4, F8, T4, T6, C4, P4) residing electrodes. Frequencies between 1 to 10 Hz were analysed to avoid muscle and motion artifact. Topoplots were shown for illustration of findings (FieldTrip toolbox [10]).

## Sleep staging, arousal scoring, and EEG pruning

Two trained members of the research team (one a clinician) independently reviewed each video-EEG recording and performed sleep staging and scoring of EEG arousals according to the American Academy of Sleep Medicine (AASM) guidelines [11]. 10-minute continuous EEG samples of wake, stage N2, and deep sleep (N3) were selected from each recording. Stage N2 sleep was identified by the presence of sleep spindles and K complexes and selected to represent light sleep, and deep sleep by epochs with >20% delta rhythms. It was not possible to reliably select samples of REM sleep as no electrooculography (EOG) or electromyography (EMG) data were available. The number of arousals was counted during the available EEG from one night of sleep (excluding the first night, where possible, to avoid first night effect) [12] and divided by the total number of hours of sleep-EEG (2–4 hours due to pruning of historical EEGs), then expressed as a number per hour of sleep. HEs were also identified during review of video-EEG and characterised independently by two members of the research team (one a clinician) for clinical features. For each HE, 5-minute epochs were selected by extracting the EEG 5-minutes pre-ictally (immediately before HE), ictally (from the clinical onset of the

HE), post-ictally (immediately after), as well as a baseline epoch was selected starting one hour prior to the HE (where possible).

## Statistical analysis

Data were tested for normal distribution and equal variance. Analysis of spectral power during sleep was performed using non-parametric Kruskal-Wallis test with factors of sleep stage (wake, light sleep, deep sleep), frequency band (delta, alpha, beta) and disease state (AHC and epilepsy control), with post-hoc Dunn's evaluation. Ictal spectral power during HE within the AHC group was performed using two-way ANOVA with factors of hemisphere and event period (baseline, pre-ictal, ictal, sleep post-ictal), with Holm-Šídák post-hoc evaluation. An unpaired t-test was used to test the significance of the difference between the number of arousals in the AHC and control groups. The significance threshold was defined as $p < 0.05$. Statistical analyses were performed using RStudio (Version 1.4.1103, RStudio Team 2021) or SigmaPlot (Systat Software, Inc).

## Results

### Clinical features

Five patients (all female) with AHC were included in the cohort having undergone video-telemetry as part of clinical care. Mean age at last follow up was 34 years (SD ± 4.5, range 29–41 years). Four of the five patients had *de novo* heterozygous variants in *ATP1A3*; one patient had a clinical diagnosis of AHC with no known genetic cause (Patient 5). Four of the five patients had a diagnosis of epilepsy (Patients 2, 3, 4, and 5) and two were drug-refractory; one patient had no known history of seizures (Patient 1). All five patients had evidence of developmental delay; intellectual difficulties were present in four patients (excluding Patient 5). MRI investigations revealed changes consistent with cerebellar atrophy in Patients 1, 2, and 5, left hippocampal sclerosis in Patient 3, and no abnormalities in Patient 4, who had a vagus nerve stimulator device *in situ* that prevented further MRIs from the age of 10 years. Additional clinical characteristics of each patient can be seen in Table 1.

### Video telemetry

VT (duration range 24–96 hours) was performed at a median age of 25.5 years (SD ± 4, range 24–34 years) for the AHC group and revealed abnormalities in all patients (see Table 1). Non-epileptic paroxysmal episodes were recorded in all five patients, including HEs (Patients 2 and 3), episodes of dystonia (Patients 2, 3, and 4), episodes of reduced awareness (Patients 2, 3 and 4), abnormal eye movements (Patients 1, 2, 3, and 4), and one episode of altered sensation (Patient 5). None of these episodes were accompanied by any EEG change detected by visual inspection.

Typical cyclic sleep architecture was not observed in Patients 1, 2, 3, and 4. In Patient 5, only 2 hours of sleep-EEG were available due to pruning of the recording, so it was not possible to appreciate typical sleep architecture. In all five patients the sleep-EEG was interrupted by frequent arousals. In Patient 1 arousals were occasionally accompanied by episodes of paroxysmal eye flickering and one of the seizures in Patient 4 was preceded by an arousal from sleep. In all three patients who had interictal epileptiform activity during wakefulness, this was more pronounced during sleep (Patients 2, 3, and 4), and in Patient 4 the sleep-EEG was further interrupted by brief periods of attenuation. The AHC group had a mean of 19 arousals per hour (SD ± 5.3, range 11–21.5), which was 98% more than the mean of 9.6 arousals per hour (SD ± 3.1, range 5.8–13.3) observed in the control group. VT was carried out at a median age

**Table 1. Clinical characteristics of AHC group.**

| Patient | *ATP1A3* Mutation | Age at onset of AHC (m) | Age of Clinical Diagnosis (y) | Age at VT (y) | MRI | Cardiac Findings | Interictal EEG | Frequency of Paroxysmal Events at VT | Arousals (per hour)[E] | Treatment at VT |
|---|---|---|---|---|---|---|---|---|---|---|
| 1 | p. Gly947Arg c.2839G>A | < 12 | 4 | 34 | 18y: unremarkable 34y: mild volume loss in the vermis | Paroxysmal repolarisation abnormalities (ECG) | Occasional generalised, bilateral bursts of theta[D] | Unclear (infrequent hemiplegic episodes) | 22 | Baclofen |
| 2 | p.Ser811Pro c.2431T>C | < 2 | < 1 | 26[B]; 31[C] | 22y: unremarkable 24y: unremarkable 25y: slight volume loss in the vermis | Dynamic changes with repolarisation and conduction abnormalities (ECG); mild mitral valve prolapse with mild regurgitation (echocardiogram) | Diffuse underlying theta transients[B]; runs of sharpened theta, bursts of bilateral, anterior dominant sharpened slow[C] | Focal seizures (4-5/month)[B]; hemiplegic episodes (every other day)[B,C]; Focal and GTCS (frequency unclear)[C] | 24 | Phenytoin, topiramate, baclofen, flunarizine[B]; phenytoin, topiramate, flunarizine, omeprazole[C] |
| 3 | p. Ser137Phe c.410C>T | < 1 | 6 | 25 | 5y: left hippocampal atrophy 23y: severe left hippocampal sclerosis | ECG normal; implantable cardiac loop device revealed episodes of asystole and a cardiac pacemaker was implanted (25y) | Diffuse slowing; paroxysmal anterior dominant, generalised spike and slow wave complexes | GTCS (4-6/ year); hemiplegic episodes (2-3/ week) | 21.5 | Carbamazepine, flunarizine, pizotifen, loratadine |
| 4[A] | p. Glu815Lys c.2443G>A | < 2 | 20 | 24 | 10y: unremarkable | Dynamic changes: repolarisation abnormalities previously detected but most recent prolonged ECG was unremarkable; Intermittent sinus tachycardia | Right anterior dominant delta and frequent spikes and notched delta, occasional spreading to left | Focal seizures (1-2/week); hemiplegic episodes (daily) | 11 | Levetiracetam, clobazam, phenytoin, lacosamide, pregabalin, flunarizine |
| 5 | *ATP1A3* negative | < 4 | 12–14 | 25 | 19y: marked cerebellar atrophy | Sinus tachycardia and microvascular angina (ECG) | Independent, bihemispheric paroxysmal theta | Possible seizures (frequency unclear); hemiplegic episodes (3-4/ week) | 16.5 | Flunarizine, pizotifen, baclofen |

[A]: Deceased;

[B]: First VT;

[C]: Second VT;

[D]: Patient does not have epilepsy;

[E]: Total number of arousals during available sleep EEG/number of hours counted.

of 23.5 years in the control group (n = 12, SD ± 9.6, range 17–49 years). An unpaired *t* test revealed that this increased number of arousals in the AHC group relative to the epilepsy control group was statistically significant (*p* = 0.0003).

## Clinical features and spectral analysis of hemiplegic episodes

Eleven HEs were identified on review of video-EEG in Patients 2 (six events) and 3 (five events). Two HEs in Patient 3 were excluded from all further analyses due to excessive artifact

**Table 2. Clinical and EEG features and lateralisation of hemiplegic episodes.**

| Patient | Event | Semiology | Clinical Lateralisation | Visible EEG Change | Lateralisation of Spectral Power** |
|---------|-------|-----------|------------------------|-------------------|-----------------------------------|
| 2 | 1 | Dystonic posturing of L arm; head slumps to L; facial movements throughout | Unclear | None | L |
| | 2 | Head slumps to L; R arm postures; facial grimacing and swallowing; L hand and arm unaffected | R | None | L |
| | 3 | No movement throughout | Unclear | None | R |
| | 4 | Unknown* | Unclear | None | L |
| | 5 | Unknown* | Bilateral | None | R |
| | 6 | Unknown* | Bilateral | None | L |
| 3 | 1 | Head turns to R then L; both upper limbs postured and stiff; grips with L hand (?hyperreflexia) | Unclear | None | L |
| | 2 | Nystagmus; unresponsive; swallowing; L hand and arm unaffected; dystonic posturing of R arm; eye blinking | R | None | L |
| | 3 | Slumps to L; swallowing; makes fists with both hands | Unclear | None | R |

L = left; R = right;

* = video not available;

** = as visible on topoplots (FieldTrip Toolbox).

or no video available. Video was not available for a further three events in Patient 2, but the EEG epochs were included for further analyses as a parent was present at the time and identified the event; of these three events, two were reported as bilateral by a parent at the time of the event. Table 2 shows the clinical features and EEG lateralisation of each of the HEs. Clear lateralising signs were identified clinically in two of the events on review of the videos, and electrographic lateralisation of z-scored low frequency (1–10 Hz) power was evident following spectral analysis in the pre-event (5 minutes immediately preceding the clinical onset) and event epochs of all nine events analysed. Example topoplots (sensor level frequency power representations) demonstrating spectral changes at baseline and throughout one event from each patient are shown in Fig 1. Topoplots demonstrating spectral changes for all nine events can be seen in S1 Fig. Spectral power values were divided by hemisphere with increased activity visible on the channel topographic plot versus hemisphere with reduced activity. A two-way ANOVA with factors of hemisphere and event period demonstrated a significant difference in z-scored low frequency power between affected and non-affected hemisphere (F = 3.39, $p = 0.034$), driven by changes during the pre-event and event epochs (Holm-Šídák *post hoc* test: $t = 2.65$, $p = 0.014$ and $t = 2.588$, $p = 0.016$ respectively). The mean ± SEM Fourier transformed z-scored power values for each hemisphere in each epoch are shown in Table 3.

## Spectral analysis of sleep and wake

Fig 2 shows the relative power for delta, alpha, and beta frequency bands during wakefulness, light sleep, and deep sleep in AHC and controls. A significant difference between the AHC and control groups was found in the wake stage only (Kruskal-Wallis test: F = 4.996, $p = 0.011$). *Post hoc* analysis with a Dunn's test revealed this was driven by changes within the delta ($p = 0.025$) and beta ($p = 0.017$) frequency bands, with 22% greater proportion of delta power in AHC versus controls and 26% more beta power in controls versus AHC during wake.

## Discussion

Disrupted sleep is a newly identified clinical feature in children with AHC [8]. Here, we show that this persists into adulthood, with sleep interrupted by frequent arousals in five adults with

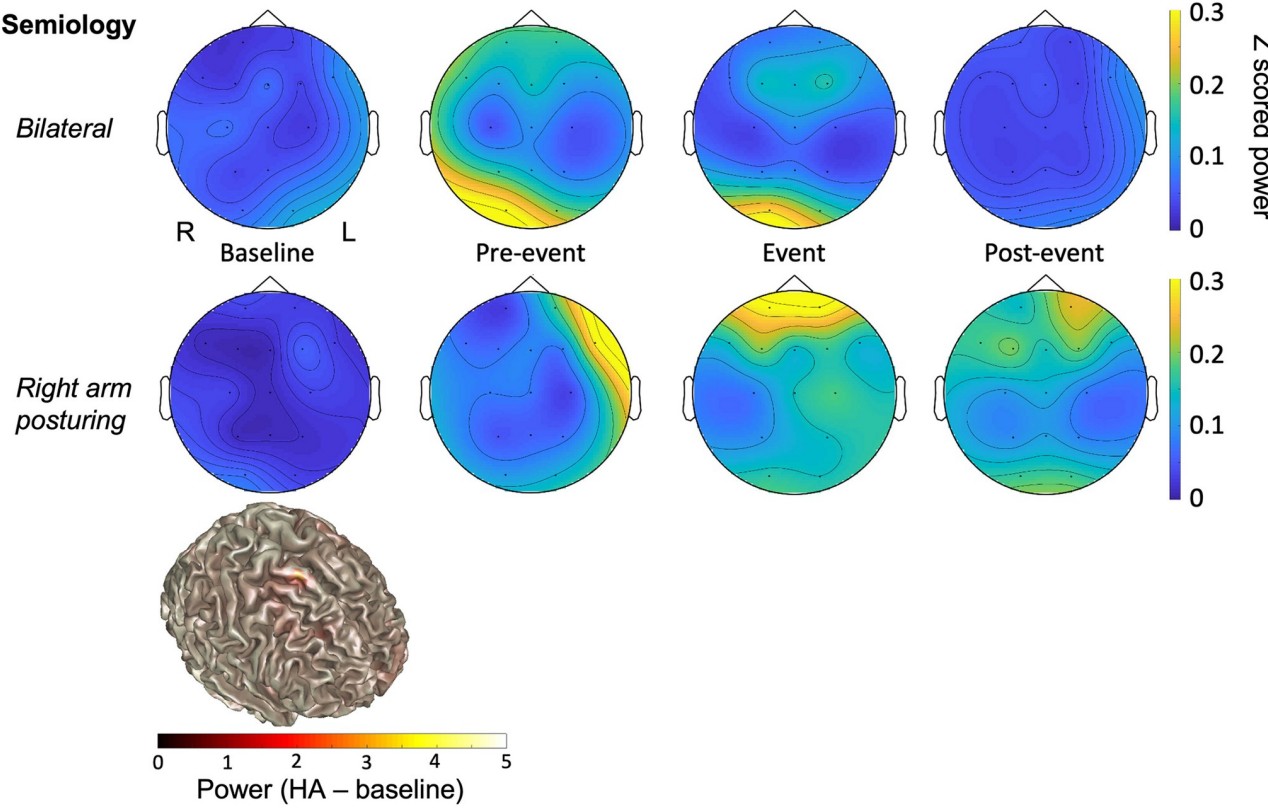

**Fig 1. Example Topoplots for each patient throughout a hemiplegic episode.** A Topographic distribution of normalised EEG power between 1 and 10 Hz at baseline, pre-event (5 minutes immediately preceding clinical onset), event (5 minutes during hemiplegic episode) and post-event (5 minutes immediately following clinical offset) epochs from one hemiplegic episode from each patient. The colour bar represents differences in normalised power (arbitrary units), with increased power represented by yellow colours and decreased power represented by blue powers. R = Right; L = Left. **B** EEG source model of low frequency power using beamforming of pre-event minus baseline conditions.

AHC. Additionally, we demonstrate a neurophysiological lateralisation of ictal power in 9 HE which occurred in two patients, detectable with scalp EEG despite no electrographic epileptiform activity, and present prior to the onset of hemiplegic episodes before resolving post-ictally.

Spectral analysis of EEG epochs throughout HEs revealed a hemispheric lateralisation of EEG power across a low frequency range (1–20 Hz) that was found to be significantly different during the event and up to 10 minutes preceding the event. For the two events with a clear clinical lateralisation, the increase in EEG power was evident contralateral to the clinically affected side. This could reflect a contralateral hemispheric slowing of EEG, which has been demonstrated in benign nocturnal alternating hemiplegia of childhood, a distinct condition

**Table 3. Mean ± SEM z-Scored hemispheric power at baseline and throughout event.**

| Hemisphere | Baseline | Pre-Event | Event |
|---|---|---|---|
| Hemisphere with Increased Activity* | 0.073 ± 0.003 | 0.266 ± 0.08 | 0.21 ± 0.06 |
| Hemisphere with Reduced Activity* | 0.062 ± 0.007 | 0.186 ± 0.06 | 0.196 ± 0.07 |

Data represents mean ± SEM fourier transformed normalised power.

* = Visible on topoplot (FieldTrip toolbox).

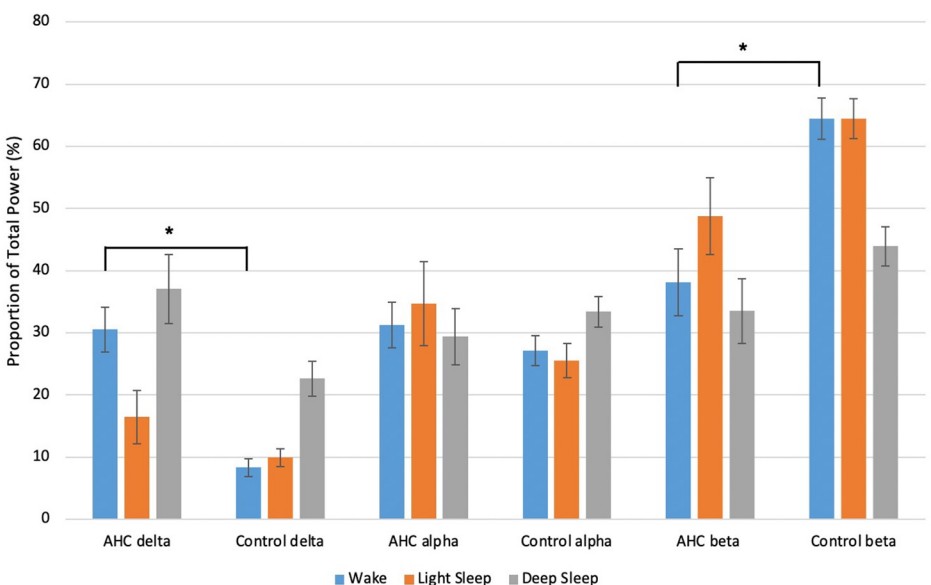

**Fig 2. Proportion of delta, alpha, and beta power in each stage for AHC and control groups.** Data represents mean ± SEM proportion of relative power for each frequency band in wake, light sleep, and deep sleep in AHC and control groups. Kruskal-Wallis with Dunn's *post hoc* test: * $p < 0.05$.

from AHC with similarities in presentation [13,14]. All but two of the ten HEs captured on VT in two patients in the present study were complicated by additional paroxysmal features, such as dystonia, which made it impossible to definitively lateralise the event clinically. This highlights an under-reported complexity to HEs and a need for a more representative clinical description, particularly in adults with AHC, for the majority of the literature characterises hemiplegia in children with AHC. Clinical manifestations are complex in AHC [15], and their complexity may increase in adulthood alongside progressive cerebral and cerebellar atrophy and correlate with clinical severity [16]. Indeed, both patients for whom HEs were captured had progressive atrophy evident on repeat MRI investigations.

Whilst we note the complexity of the clinical presentation of the HEs, and the small number of observations, a hemispheric spectral change preceding the event was observed in all HEs analysed, including during bilateral plegic episodes. This may provide an initial suggestion, needing validation in larger studies, that the underlying neural mechanisms involved at the initiation of HEs are unilateral, regardless of the clinical presentation. One potential mechanism could involve a dysfunction of cortical-subcortical networks in the aetiology of hemiplegia in AHC, with a potential involvement of basal ganglia structures. Cortical spreading depression, which involves a slowly propagating wave of depolarization followed by suppression of brain activity, has been associated with disruption of $Na^+/K^+$ gradient(Haglund et al., 1990) and suggested as a potential mechanism underlying AHC(Hunanyan et al., 2015). Our findings support the hypothesis that an initial depolarisation change starts on one hemisphere and is then followed by bilateral changes in brain activity. If reproduced in larger AHC cohorts, the spectral changes present minutes before the event might be proposed as a predictive tool for HEs, which would benefit both patient safety and care provided. Additionally, this finding could act as a potential neurophysiological biomarker for future therapies.

As there is a well characterised effect of sleep in resolving HEs, we carried out spectral analysis of EEG during sleep and wakefulness to determine if the restorative benefit of sleep in

AHC is accompanied by spectral changes. Spectral analysis of EEG revealed significant differences during wake periods between patients with AHC and epilepsy control patients. In particular, there was a 22% increase in the proportion of delta power and a 26% decrease in the proportion of beta power during wake in the AHC group relative to controls. Although there was a trend of increased delta power proportion during deep sleep between AHC and epilepsy control patients visible in Fig 2, statistical analysis of sleep-EEG during light and deep sleep did not reveal any significant differences between the two groups.

With regards to the EEG spectra, the low frequency power shift seen in the people with AHC during wakefulness could represent global cerebral dysfunction [3], as is seen in other conditions, such as epilepsy and Angelman syndrome [17]. However, it is not possible to exclude that the changes in low frequency power seen in the people with AHC could be underpinned by MRI abnormalities, which were present in four out of five patients in the present cohort. Whilst no differences were found between the groups during light sleep and deep sleep, this could be due to the small sample size of the present study. The lack of a significant difference in power during deep sleep, where a trend of increased delta power was observed in the people with AHC, was particularly surprising. Alternatively, this could be underpinned by changes which clinical scalp EEG is insensitive to, such as altered frequency coupling or network activity.

In this small adult cohort, we observed that sleep architecture was disrupted in all five adults with AHC, with frequent arousals (5 patients), epileptiform activity (3 patients), paroxysmal eye flickering (1 patient), and seizures (1 patient) occurring from sleep. There were 98% more arousals per hour of sleep in the AHC group compared to epilepsy controls, and the number of arousals per hour for every patient with AHC was higher than the control mean. Whilst sleep disruption is common in epilepsy [18], our study suggests that sleep in AHC is disrupted by a separate mechanism; indeed, frequent arousals were observed in Patient 1, who did not have epilepsy. The number of arousals observed in the present cohort was 27% greater than the arousal index previously reported in children with AHC [8]. We do not know whether sleep disruption and the restorative effect of sleep on HEs may change with age in AHC, but this is an important aspect to study and ongoing studies on the natural history of the disease may help to clarify the variation of sleep features over time.

There were limitations to the present study. In particular, the small sample size of the AHC group and retrospective study design limit the power and generalisability of results. Patients with well-controlled epilepsy were selected for the epilepsy control group, while 50% of the patients with AHC with epilepsy had drug-resistant seizures. Thus, it is not possible to exclude that the co-existent epilepsy contributed to the aetiology of sleep disruption in the AHC group. However, we excluded any epochs containing clinical or electrographic seizures or interictal epileptiform abnormalities for power analysis. Another potential source of confounding was the complex polytherapy, mostly comprising antiseizure medications, that both the patients with AHC and the epilepsy control group received. This could have affected the EEG power and arousal data. As the video-EEGs were carried out over the period of 9 years there was variability in the methodology used and limited data for the historical recordings. Additionally, as respiratory data were not recorded at the time of video-EEG, it was not possible to detect sleep apnoeas, which have been described in children with AHC [8]. There is therefore a need for polysomnography (PSG) studies in adults with AHC. A systematic clinical evaluation of any sleep disorders was not possible either, due to the retrospective study design. Finally, only one *ATP1A3*-negative patient was included and could therefore have had a different pathophysiology to the other four patients. However, a clinical diagnosis according to the Aicardi criteria was made [7].

Here, we demonstrate that sleep is disrupted in adults with AHC, as it is in children, further expanding the complex phenotypic spectrum and suggesting a need for routine PSG investigations in the clinical care of these patients, particularly given the risk of nocturnal seizures and sudden unexpected death in epilepsy (SUDEP). In addition, our results show that there are lateralised electrographic spectral changes associated with hemiplegic episodes in AHC that can be detected up to ten minutes prior to event onset. This work could go on to inform novel strategies in the management of AHC, and possibly shed light on the underlying physiology of other conditions associated with mutations in *ATP1A3*, such as hemiplegic migraine.

## Supporting information

**S1 Fig. Topoplots for all hemiplegic episodes.** Topographic distribution of normalised EEG power between 1 and 10 Hz at baseline, pre-event (5 minutes immediately preceding clinical onset), event (5 minutes during hemiplegic episode) and post-event (5 minutes immediately following clinical offset) epochs from all hemiplegic episodes analysed. From top to bottom: Patient 3 events 1–3 (left) and Patient 2 events 1–6 (right) and. Beside each event is a clinical lateralisation of attack (semiology). The colour bar represents differences in normalised power (arbitrary units), with increased power represented by yellow colours and decreased power represented by blue powers. R = Right; L = Left.
(TIF)

**S1 Table. Age, gender and treatment at the time of EEG of the epilepsy control cohort.**
(DOCX)

**S1 File. Raw EEG of hemiplegic episodes.** Each of the episodes from the two patients has a separate 'pre' (5 minutes immediately preceding clinical onset), 'event'(5 minutes during hemiplegic episode) and 'post' (5 minutes immediately following clinical offset) EEG file in addition to baseline 'con_wake' control file.
(ZIP)

## Acknowledgments

The authors would like to thank all participants and their families for taking part in this research.

## Author Contributions

**Conceptualization:** Matthew C. Walker, Sanjay M. Sisodiya, Simona Balestrini, Umesh Vivekananda.

**Data curation:** Josephine Poole, Sara Zagaglia, Rita Demurtas, Fiona Farrell.

**Formal analysis:** Josephine Poole, Simona Balestrini, Umesh Vivekananda.

**Investigation:** Fiona Farrell.

**Methodology:** Matthew C. Walker, Sanjay M. Sisodiya, Simona Balestrini, Umesh Vivekananda.

**Supervision:** Matthew C. Walker, Sanjay M. Sisodiya, Simona Balestrini, Umesh Vivekananda.

**Writing – original draft:** Josephine Poole.

**Writing – review & editing:** Josephine Poole, Rita Demurtas, Matthew C. Walker, Sanjay M. Sisodiya, Simona Balestrini, Umesh Vivekananda.

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
