## [Decision Letter · Decision Letter 0]

26 Jul 2022

PONE-D-22-12955ALTERNATING HEMIPLEGIA OF CHILDHOOD: AN ELECTROCLINICAL STUDY OF SLEEP AND HEMIPLEGIAPLOS ONE

Dear Dr. Vivekananda,

Thank you for submitting your manuscript to PLOS ONE. After careful consideration, we feel that it has merit but does not fully meet PLOS ONE’s publication criteria as it currently stands. Therefore, we invite you to submit a revised version of the manuscript that addresses the points raised during the review process.

ACADEMIC EDITOR: Please do follow very useful comments and directions provided by the reviewers. Justify and explain the current sample size. 

We look forward to receiving your revised manuscript.

Kind regards,

Dragan Hrncic

Academic Editor

PLOS ONE

Journal Requirements:

“The authors would like to thank all participants for taking part in this research. S.M.S. and S.B. are funded by the Epilepsy Society, who supported this work. S.B. receives additional support from the Muir Maxwell Trust. Union Chimique Belge (UCB) provided financial support for J.P., S.Z., and R.D. The Medical Research Council, Epilepsy Research UK, the Academy of Medical Sciences, and the National Institute for Health Research (NIHR) Biomedical Research Centres (BRC) funding scheme supported U.V. Part of this work was carried out at University College London Hospitals, which receives a proportion of funding from the NIHR BRC funding scheme, who also support M.C.W.  ”

Reviewers' comments:

Reviewer's Responses to Questions

**Comments to the Author**

1. Is the manuscript technically sound, and do the data support the conclusions?

Reviewer #1: Yes

Reviewer #2: Yes

2. Has the statistical analysis been performed appropriately and rigorously? 

Reviewer #1: Yes

Reviewer #2: N/A

3. Have the authors made all data underlying the findings in their manuscript fully available?

Reviewer #1: No

Reviewer #2: Yes

4. Is the manuscript presented in an intelligible fashion and written in standard English?

Reviewer #1: Yes

Reviewer #2: Yes

5. Review Comments to the Author

Reviewer #1: This retrospective study assessed EEG spectral analysis in 5 adults with alternating hemiplegia of childhood (AHC) and

12 age-/gender-matched epilepsy controls. The data show two main statistically significant results: hemispheric differences in pre-ictal and ictal spectral power during AHC hemiplegic episodes; reduced beta power and increased delta power during wakefulness in AHC patients, in comparison with controls.

Data are clearly reported, statistically analyzed and appropriately discussed.

Both AHC patients and epilepsy controls were taking drugs at the time of the study. This represents an insurmountable limit of the paper. I think that the Authors should specify in the text or in a supplementary file which drugs the epilepsy controls were taking. Also in the Table 1 another column could contain the drugs taken by the AHC patients.

Reviewer #2: This paper examines the role of sleep in the rare disorder of Alternating Hemiplegia of Childhood in cases studied in adult life. AHC is a fascinating disorder where seizures are frequent but ill-understood other paroxysmal phenomena occur, most prominently hemiplegia, that resolves with sleep.

While sleep has been studied in this disorder, the novel aspect of this paper is the examination of adult subjects and the more detailed spectral analysis performed on hemiplegic attacks, although only 2 of the 5 cases had these. The clinical features of the 5 cases are typical of AHC. The description of the telemetry recordings of the events is useful although this was done retrospectively and eye movement monitors were not used preventing scoring of REM sleep.

Key findings were lateralization on EEG during the events where this was interpretable and spectral clots documenting the hemispheric asymmetry during events. A change in spectral power between AHC and control groups was found in the awake state only, the significance of this is a little unclear.

It would be useful if the lateralization could be more clearly described. For example, in table 3, the hemispheres are divided up in to those with “increased activity” and “reduced activity” but it is not immediately clear whether which hemisphere is relevant to the hemiplegic limb. Similarly, with the topographic maps shown on figure 1, it is not really clear what side of the body was paralysed in relationship to the evolution of the spectral power pre and post event and this should be clarified.

Under ‘significance’ the authors state that disrupted sleep is related to SUDEP. It is quite unclear what the data in this paper adsd to the understanding of SUDEP, in general or in AHC and it is nor even properly discussed. I recommend this be deleted

Overall this represents a modest but useful addition to the literature on this fascinating syndrome.

6. PLOS authors have the option to publish the peer review history of their article (what does this mean?). If published, this will include your full peer review and any attached files.

Reviewer #1: No

Reviewer #2: No

---

## [Author Response · Author response to Decision Letter 0]

5 Sep 2022

Dear Editors,

Thank you for sending our manuscript for peer review, and for giving us the opportunity to address the reviewers’ comments. Below we provide detailed answers to the comments made by the reviewers, and tracked all the changes in the manuscript. In addition, we addressed the journal requirements as requested. 

Reviewer #1: We are grateful that the reviewer found the data clearly reported, statistically analyzed and appropriately discussed.

Both AHC patients and epilepsy controls were taking drugs at the time of the study. This represents an insurmountable limit of the paper. I think that the Authors should specify in the text or in a supplementary file which drugs the epilepsy controls were taking. Also in the Table 1 another column could contain the drugs taken by the AHC patients.

We thank the reviewer for this important point. We have now added treatment data for both patients and epilepsy controls in Table 1 and Supporting Table 1, respectively. We also wrote in the discussion (page 9):

‘Another potential source of confounding was the complex polytherapy, mostly comprising antiseizure medications, that both the patients with AHC and the epilepsy control group received. This could have affected the EEG power and arousal data.’

Reviewer #2: We agree with the reviewer that AHC is a fascinating disorder where seizures are frequent but ill-understood other paroxysmal phenomena occur, most prominently hemiplegia, that resolves with sleep.

While sleep has been studied in this disorder, the novel aspect of this paper is the examination of adult subjects and the more detailed spectral analysis performed on hemiplegic attacks, although only 2 of the 5 cases had these. The clinical features of the 5 cases are typical of AHC. The description of the telemetry recordings of the events is useful although this was done retrospectively and eye movement monitors were not used preventing scoring of REM sleep.

Key findings were lateralization on EEG during the events where this was interpretable and spectral clots documenting the hemispheric asymmetry during events. A change in spectral power between AHC and control groups was found in the awake state only, the significance of this is a little unclear.

It would be useful if the lateralization could be more clearly described. For example, in table 3, the hemispheres are divided up in to those with “increased activity” and “reduced activity” but it is not immediately clear whether which hemisphere is relevant to the hemiplegic limb. Similarly, with the topographic maps shown on figure 1, it is not really clear what side of the body was paralysed in relationship to the evolution of the spectral power pre and post event and this should be clarified.

We thank the reviewer for this important distinction. Even with experienced neurophysiologists reviewing the attacks, due to the complex nature of them, it was challenging to definitively assign clinical lateralisation to the majority. Indeed normal hemi-function was demonstrated in two events where we can be certain about lateralisation. In these two cases the corresponding lateralisation of spectral power (i.e. contralateral to clinical semiology) was demonstrated. What is interesting is that nearly all attacks demonstrated prior spectral lateralisation irrespective of clinical semiology, which argues the possibility that attack onset has a focus. For this reason, we used increased/reduced spectral activity as a comparator in order to permit statistical analysis. For ease of understanding we have outlined each of the attacks in Table 2, and amended Figure 1 and Supplemental Figure 1 to incorporate clinical semiology when possible. 

Under ‘significance’ the authors state that disrupted sleep is related to SUDEP. It is quite unclear what the data in this paper adds to the understanding of SUDEP, in general or in AHC and it is nor even properly discussed. I recommend this be deleted.

We have now removed the statement on SUDEP, as suggested.

---

## [Decision Letter · Decision Letter 1]

19 Sep 2022

ALTERNATING HEMIPLEGIA OF CHILDHOOD: AN ELECTROCLINICAL STUDY OF SLEEP AND HEMIPLEGIA

PONE-D-22-12955R1

Dear Dr. Vivekananda,

We’re pleased to inform you that your manuscript has been judged scientifically suitable for publication and will be formally accepted for publication once it meets all outstanding technical requirements.

Kind regards,

Prof. Dr. Dragan Hrncic, MD, PhD

Academic Editor

PLOS ONE

Additional Editor Comments (optional):

Reviewers' comments:

Reviewer's Responses to Questions

**Comments to the Author**

1. If the authors have adequately addressed your comments raised in a previous round of review and you feel that this manuscript is now acceptable for publication, you may indicate that here to bypass the “Comments to the Author” section, enter your conflict of interest statement in the “Confidential to Editor” section, and submit your "Accept" recommendation.

Reviewer #1: All comments have been addressed

Reviewer #2: All comments have been addressed

2. Is the manuscript technically sound, and do the data support the conclusions?

Reviewer #1: Yes

Reviewer #2: Partly

3. Has the statistical analysis been performed appropriately and rigorously? 

Reviewer #1: Yes

Reviewer #2: N/A

4. Have the authors made all data underlying the findings in their manuscript fully available?

Reviewer #1: Yes

Reviewer #2: Yes

5. Is the manuscript presented in an intelligible fashion and written in standard English?

Reviewer #1: Yes

Reviewer #2: Yes

6. Review Comments to the Author

Reviewer #1: The Authors addressed all the reviewers' comments and queries, improving the quality of their paper.

Reviewer #2: My concerns have been addressed. My concerns have been addressed. My concerns have been addressed. My concerns have been addressed.

7. PLOS authors have the option to publish the peer review history of their article (what does this mean?). If published, this will include your full peer review and any attached files.

Reviewer #1: No

Reviewer #2: No

---

## [Editor Report · Acceptance letter]

21 Sep 2022

PONE-D-22-12955R1 

Alternating Hemiplegia of childhood: an electroclinical study of sleep and Hemiplegia 

Dear Dr. Vivekananda:

I'm pleased to inform you that your manuscript has been deemed suitable for publication in PLOS ONE. Congratulations! Your manuscript is now with our production department. 

Kind regards, 

on behalf of

Professor Dragan Hrncic 

Academic Editor

PLOS ONE